# Full-Color Imaging System Based on the Joint Integration of a Metalens and Neural Network

**DOI:** 10.3390/nano14080715

**Published:** 2024-04-19

**Authors:** Shuling Hu, Ruixue Shi, Bin Wang, Yuan Wei, Binzhi Qi, Peng Zhou

**Affiliations:** 1School of Instrumentation and Optoelectronics Engineering, Beihang University, Beijing 100191, China; hulxi@buaa.edu.cn (S.H.); qibinzhi@buaa.edu.cn (B.Q.); 15204071233@163.com (P.Z.); 2Institute of Microelectronics of the Chinese Academy of Sciences, Beijing 100029, China; 3Photonic Institute of Microelectronics, Wenzhou 396 Xingping Road, Longwan District, Wenzhou 100029, China; weiyuan@pime.ac.cn

**Keywords:** metalens, end-to-end joint neural network, high-quality image system

## Abstract

Lenses have been a cornerstone of optical systems for centuries; however, they are inherently limited by the laws of physics, particularly in terms of size and weight. Because of their characteristic light weight, small size, and subwavelength modulation, metalenses have the potential to miniaturize and integrate imaging systems. However, metalenses still face the problem that chromatic aberration affects the clarity and accuracy of images. A high-quality image system based on the end-to-end joint optimization of a neural network and an achromatic metalens is demonstrated in this paper. In the multi-scale encoder–decoder network, both the phase characteristics of the metalens and the hyperparameters of the neural network are optimized to obtain high-resolution images. The average peak-signal-to-noise ratio (PSNR) and average structure similarity (SSIM) of the recovered images reach 28.53 and 0.83. This method enables full-color and high-performance imaging in the visible band. Our approach holds promise for a wide range of applications, including medical imaging, remote sensing, and consumer electronics.

## 1. Introduction

Today, imaging systems are widely used in medical instruments, wearable devices, smartphones, and so on. However, modern imaging systems usually consist of multiple optical elements to overcome geometric aberrations. The introduction of additional components, such as lenses, mirrors, or prisms, increases the overall weight and volume of the system, which may limit the application of imaging systems [1,2]. How to build a miniaturized imaging system and maintain high performance has become a hot topic in the industry and academia.

Nowadays, by manipulating the geometrical parameters of the subwavelength elements, such as the size, shape, and orientation, metasurfaces can modulate the polarization [3], amplitude [4,5], and phase [6] of incident light to achieve the desired functionality. As one planar optical device derived from metasurfaces, metalenses hold tremendous potential in the field of optical imaging. Different from traditional lenses [7], metalenses do not rely on changes in the thickness of the constituent structures to accumulate phase, but directly modulate the phase of the incident light. The emergence of metalenses addresses the issue of bulky volume associated with conventional optical lenses, aligning with the trends in integration and miniaturization [8,9,10] and offering various functionalities. However, the phase discontinuity of metalens can also lead to image distortion and blur. The design of achromatic metalens is still limited by large aperture and low Fnumber [11,12].

In recent years, various inverse designs have been proposed for nanomaterials and metelenses [13,14,15]. Sensong An [16] established a forward spectral prediction tensor neural network to predict the transmission spectra of meta-atoms with different structures. Zhaocheng Liu [17] demonstrated the feasibility of using an unsupervised learning system [18] to reverse-design nanophotonics. But these methods belong to the forward design metalens element structure. Meanwhile, existing end-to-end optimization frameworks in meta-optics [19,20,21] cannot optimize the final full-color image quality. They typically rely on intermediate metrics such as spot intensity.

Computational imaging opens up new directions for improving imaging quality [22,23,24,25,26,27,28]. The integration of metasurface optics and deep learning methods has significantly advanced high-quality images [29,30,31,32,33,34]. In 2019, Vincent Sitzmann et al. [23] designed an end-to-end optimization system for dispersion-compensated wide field depth and super-resolution imaging, integrating optical and image processing components. The system is a fully differentiable model that jointly optimizes the effective refractive index of diffractive optical elements (DOEs) and image processing parameters, but the modulation of DOEs is limited by phase. In 2022, Zeqing Yu et al. [35] utilized a U-Net pre-processing model and incorporated both metalens and computationally generated holography into one imaging system. In 2022, Qiangbo Zhang et al. [36] proposed a snapshot hyperspectral imaging system based on metalens, achieving joint optimization of the metalens and image processing. However, both methods are designed for polarization-sensitive metalenses.

In this paper, a high-quality imaging system is proposed by jointly optimizing a neural network and a polarization-insensitive achromatic metalens. The imaging system, outlined in Figure 1, employs both forward and backward propagation networks for image reconstruction. During forward propagation, the ground truth is convolved with the point spread function (PSF) of the metalens, and noises are added to generate the sensor image. Subsequently, the neural network reconstructs the sensor image. The loss function is computed by comparing the reconstructed images with the ground truth. In the process of backward propagation, the neural network and metalens parameters are optimized to minimize the loss function. Our method provides a new method for full-color imaging using a polarization-insensitive metalens. The polynomial phase factor makes the design more flexible to help achieve an achromatic metalens, and the multi-scale neural network makes the image feature extraction more comprehensive and conducive to image reconstruction. This approach enhances high-quality image recovery by co-optimizing the front-end optics and back-end recovery network.

## 2. Theoretical Analyses

The optimization process is mainly divided into the following steps. First, the metalens is constructed by selecting nanopillars according to the phase profile. The corresponding phases of nanopillars with different diameters are obtained by the finite difference time domain method. The phase profile of the metalens consists of hyperbolic and polynomial phases. Polynomial factors are optimized to design the achromatic metalens. Second, the PSF of the metalens is calculated, the ground truth is convolved with the PSF, and noises are added to create the sensor image. Third, the neural network is built and the phase factor of the metalens is optimized along with the network hyperparameters. Fourth, the loss function is minimized by comparing the loss between the reconstructed image and the ground truth.

### 2.1. Metalens Design

The phase control of a metalens can be divided into the geometric phase and propagation phase. However, since the geometric phase is sensitive to polarization, this paper focuses on the polarization-insensitive propagation phase. The propagation phase method provides phase discontinuities by altering size such as the height and diameters of symmetric unit cells. A nanopillar has structural symmetry and can be modulated by changing the duty cycle [37,38]. Silicon nitride is easy to integrate and has high transmittance in the visible spectrum. Therefore, a silicon nitride nanopillar with the propagation phase is discussed here. 

An important aspect of designing a metalens is optimizing the geometric parameters of its unit structures, which include the period, diameter, and height, to achieve phase coverage from −π to π. According to the Nyquist sampling theorem, the period P of the unit structure should satisfy P < λ/2NA [39], where NA is the numerical aperture of the metalens. To suppress higher-order diffraction, the period of the unit structure should be smaller than the wavelength of the incident wave [40,41]. However, for achromatic metalenses operating within a certain wavelength range, the period of the unit structure should be larger than the wavelength of the incident light to excite resonance of different dispersive modes [40,41]. As the height of the unit structure increases, the achievable range of phase modulation also increases. However, a higher aspect ratio of the unit structure makes fabrication more challenging. Figure 2a shows the design of the unit cell. The selected nanopillar period is 350 nm, and the height is 0.8 μm. 

In this paper, the finite difference time domain method is used for simulation. The diameter was swept from 50 to 350 nm. Figure 2c,d show the phase and transmittance of three wavelengths, respectively. Multiple −π to π periods are achieved by changing the diameter of unit cells, which means that more than one nanopillar can be selected at a certain location for the target phase. A phase library corresponding to the diameters of the nanopillars was constructed. As shown in Figure 2c,d, the transmittance remains at a high level except for the falling peaks.

The metalens phase profile in this paper is designed in the form of
(1)$$\varphi_{lens}(x,y,z,\lambda )=-\frac{2\pi }{\lambda }(\sqrt{x^{2}+y^{2}+f^{2}}-f)+{\sum_{i=0}^{8}a_{i}(\frac{x^{2}+y^{2}}{R^{2}})}^{i}$$
where the first term in the formula is the hyperbolic phase and the second term is the polynomial phase. (*x*, *y*) is the position from the center of the metalens, *f* is the focal length, *λ* is the target wavelength, and *a_i_* is the polynomial phase factor as the optimizable parameter. According to Equation (1), the phase of the metalens at each position is obtained, and the corresponding nanopillar is selected from the phase library. Traditional metalenses use the hyperboloidal phase to generate a perfect spherical wavefront [37], but its coefficients are fixed and cannot be optimized. Here, three wavelengths of 462 nm, 511 nm, and 606 nm were selected for the achromatic metalens design in the visible band. The phase factors of the three wavelengths were optimized and initialized by the particle swarm optimization (PSO) algorithm [42]. Then, the phase factors were fine-tuned by using end-to-end network imaging.

### 2.2. PSF Calculation

The imaging system is modeled as a convolution of the ground truth with the PSF. The PSF is a function used to describe the imaging performance of an optical system for a point source of light. When a point source of light is imaged through an optical system, the PSF characterizes how the light is spread out on the imaging plane because of the limitations and characteristics of the optical system. 

Scalar diffraction theory is used for imaging analysis of nano-optical elements. The metalens diffraction imaging schematic is demonstrated in Figure 3. Assuming that the amplitude of the incident light is A and the phase is *φ_d_*, when the light passes through the metalens surface, the complex amplitude after metalens modulation is written as [43]
(2)$$U(x_{0},y_{0})=A\exp(i\varphi_{d}(x_{0},y_{0}))\exp(i\varphi_{lens}(x_{0},y_{0}))$$

The light intensity received on the sensor is
(3)$$U(x,y)=\frac{1}{i\lambda z}\exp(ikz)\iint U(x_{0},y_{0})\exp(\frac{ik}{2z}[(x-x_{0}{)}^{2}+(y-y_{0}{)}^{2}])dx_{0}dy_{0}$$

The PSF is the intensity distribution of the point light source in the image plane after passing through the optical system. To obtain the PSF of the metalens, we assume that the incident light is a plane wave, and its complex amplitude is unit, then
(4)$$PSF=U(x,y)=\frac{1}{i\lambda z}\exp(ikz)\iint \exp(i\varphi_{lens}(x_{0},y_{0}))\exp(\frac{ik}{2z}[(x-x_{0}{)}^{2}+(y-y_{0}{)}^{2}])dx_{0}dy_{0}$$
(5)$$PSF\propto {\left|F\{\exp[i(\varphi_{lens}(x_{0},y_{0})+\frac{\pi }{\lambda z}(x_{0}^{2}+y_{0}^{2}))]\}\right|}^{2}$$

Only single-wavelength point source imaging is considered above, but the model can be extended to color imaging. The image formed on the sensor is a displacement invariant convolution of the ground truth and the PSF,
(6)$$I_{sensor}=\int (I_{\lambda }*P_{\lambda })d\lambda +n_{g}+n_{p}$$
where *I_sensor_* is the sensor image, *I_λ_* is the ground truth, *P_λ_* is the PSF at a certain wavelength, *n_p_* is Poisson noise, and *n_g_* is Gaussian noise. 

### 2.3. Network Architecture

In this section, a multi-scale encoder–decoder [44] based on a convolutional neural network is proposed, as illustrated in Figure 4. Here, both the multi-scale feature extraction encoder and the multi-scale decoder are fully convolutional networks.

In the encoder, a series of convolutional neural networks are employed to transform the three-channel RGB image into feature tensors. The input layer achieves resolution reduction through convolution to capture the rich feature information from the ground truth. Three different convolution kernels are 1 × 1, 3 × 3, and 5 × 5, with 15, 30, and 60 channels and correspondence to the original resolution [29], 2× downsampled resolution, and 4× downsampled resolution. At different resolutions, feature extraction is performed using residual blocks and full convolutions. The low-resolution feature maps are concatenated with the up-scale features by up-sampling. Performing feature extraction at lower resolutions may allow the network to show features at a global level, while the images extracted at the original resolution focus more on local details. Residual structures and concatenation layers are utilized to fuse different resolution features to obtain comprehensive image information.

To deal with different image resolutions, we preprocess the PSF by resizing it to 1×, 2×, and 4× downsampled resolutions. After the encoder, the image tensors are then fed into the multi-feature decoder. In the decoder, we first extract features from the original resolution using residual blocks. These features are then concatenated with the 2× downsampled resolution features. Similarly, the 2× downsampled resolution feature maps are concatenated with the 4× downsampled feature maps. After residual blocks, the 4× downsampled features are concatenated with upper-scale features using transpose convolution until the feature map returns to the original resolution. Eventually, all the feature tensors are generated a single three-channel RGB output image. This process helps the image model learn more realistic features and reconstruct high-resolution images.

### 2.4. Loss Definition

The loss function is defined as the combination of mean squared loss and perceptual losses to evaluate the deviation of the recovered image and the ground truth
(7)$$L=\lambda_{1}L_{1}+\lambda_{perc}L_{perc}$$
where the weight coefficients of *λ*_1_ and *λ_perc_* are set as 0.01 here. *L*_1_ represents the mean square error loss function. *L_perc_* is a VGG-based perceptual loss function [45].

Although the traditional mean square error loss function can obtain a high peak signal-to-noise ratio, the reconstructed image edge is too smooth. Perceptual loss learns the original graphic structure and background information by observing the combination of high and low level features extracted. Therefore, both the mean square error function and the perceived loss function are added in this paper. The perceptual loss function extracts and compares features from the output RGB image *I_out_* and the real RGB image *I_gt_* using the pre-trained VGG-19 network [45]:(8)$$L_{perc}(I_{out},I_{gt})=\sum_{b=2,3}L_{1}(\varphi_{b,2}(I_{out}),\varphi_{b,2}(I_{gt}))$$
where *φ_b_*_,2_ is the feature map extracted by the VGG-19 network at the blockb_conv_2_ layer, *I_gt_* is the ground truth, and *I_out_* is the output image.

To obtain high quality iterative images, in each iteration, the metalens phase factor and neural network system parameters should be continuously optimized to minimize the loss. Then the expression can be written as
(9)$$\{M_{\mathrm{lens}}^{*},M_{\mathrm{CNN}}^{*}\}=\mathrm{argmin}\sum_{i=1}^{N}L(I_{out},I_{gt})$$
where *N* is the number of training samples, *M_lens_* is the metalens parameter, *M_CNN_* is the network parameter, *I_gt_* is the ground truth, and *I_out_* is the reconstructed output image. After completing the training, *M_lens_* is used to design our metalens.

## 3. Results and Discussion

### 3.1. Experimental Details

For the metalens design, the focal length was set to 15 mm and the diameter was 1 mm. The distance between the metalens and sensor was set to the focal length. A deep learning platform based on TensorFlow 2.1.6 was used, and the GPU was NVIDIA P100 (Santa Clara, CA, USA) with 16 GB memory in the training and testing experiment.

The DIV2K dataset [46] was used as the training set. The DIV2K dataset consists of over 800 high-resolution images. These images are sourced from various origins and cover diverse scenes and subjects. The dataset encompasses a variety of image types, including natural landscapes, portraits, architecture, etc., to ensure robust performance evaluation of algorithms across different scenarios. Typically, the DIV2K dataset is divided into training and testing sets. The training set is utilized for model training, while the testing set is used to evaluate model performance. The DIV2K dataset is widely used to evaluate the performance of image super-resolution algorithms, both qualitatively and quantitatively. It serves as a benchmark for training and testing various super-resolution models, including deep learning-based approaches. The dataset is enhanced by flipping the training image horizontally, vertically, and both horizontally and vertically triple the number of images. The image is cut to 720 × 720 size.

The parameter optimization algorithm uses Adam optimizers (*β*_1_ = *β*_2_ = 0.9). In each optimization process, the method of alternating optimization is used to optimize the phase factor and network parameters, respectively. In each iteration, the phase is optimized 5 times with a learning rate of 0.004, and the convolutional neural network parameters are optimized 10 times with a learning rate of 0.00095. The batch size is set as two. The training was conducted for 3000 iterations, which took 9 h. Our sensor camera is the Prosilica GT2000 (Burnaby, BC, Canada) with 5.5 μm pixels and the reconstructed image resolution is 720 px × 720 px, which matches the training image size. The sensor is modeled as Gaussian noise and Poisson noise, where η_g_(x, σ_g_)~N(x, σ_g2_) is the Gaussian noise component and η_p_(x, a_p_)~P(x/a_p_) is the Poisson noise component. Where σ_g_ = 1 × 10^−5^, a_p_ = 4 × 10^−5^.

### 3.2. Results

The normalized simulated PSF is shown in Figure 5. It can be seen that the focusing effect is good at the three wavelengths of 462 nm, 511 nm, and 606 nm, and the defocusing will reduce the image quality. With the help of neural networks, the defocusing effect can be partially offset.

The image quality is quantitatively analyzed by using the peak-signal-to-noise ratio (PSNR) and structure similarity (SSIM). The PSNR and SSIM kept increasing and tended to stabilize after a certain number of iterations. The algorithm model in this section achieved good results in convergence, as shown in Figure 6. PSNR training improves after starting training. Between 0 and 500 iterations, the PSNR of the model fluctuatez greatly. On the whole, the PSNR kept rising and remained stable after 500 iterations, indicating that rapid convergence can gradually decrease.

At the same time, we also compared the metalens based on the cubic phase and hyperboloid phase. The cubic phase formulation can be written as
(10)$$\varphi (x,y)=\frac{2\pi }{\lambda }(\sqrt{x^{2}+y^{2}+f^{2}}-f)+\frac{a}{R^{3}}(x^{3}+y^{3}),$$
where (*x*, *y*) is the position of the metalens, *f* is the focal length, *λ* is the wavelength, and *R* is the metalens radius. A is the design parameter of the cubic term and is set to 86π.

The hyperboloid phase formulation can be written as
(11)$$\varphi (x,y)=\frac{2\pi }{\lambda }(\sqrt{x^{2}+y^{2}+f^{2}}-f)$$
where *λ*_0_ = 462 nm is the nominal wavelength and *f*_0_ = 15 mm is the nominal focal length. We set *f* = *f_0_*·*λ*/*λ*_0_.

Table 1 shows the PSNR and SSIM of images recovered by three methods [6,21]. The recovery data of our model is better than the other two methods. Our method achieves an average PSNR of 28.53, which is about 6 dB better than the cubic phase method and about 11 dB better than the hyperboloid phase method. The average SSIM of the output image reaches 0.83, which is about 0.2 higher than the cubic phase method and about 0.3 higher than the hyperboloid method.

The final output image results are shown in Figure 7. The first column is the ground truth. The second column is the output image reconstructed in this paper. In contrast, the third and fourth columns are the reconstructed images based on the cubic and polynomial phases, respectively. It can be seen that the quality of the second line of restored images is significantly better than that of the second and third lines, both in terms of color and detail. The artifacts, blur, and noise at the edge of the images have been effectively restored and eliminated. Although it cannot be restored to the ground truth, it still demonstrates the good recovery capability of our imaging system.

### 3.3. Discussion

Our method combines the optimization of metalens phase parameters and neural network hyperparameters through an end-to-end network to generate high-quality reconstructed images. Compared with the hyperbolic phase method and cubic phase method, our method adds a polynomial phase factor to make metalens regulation more flexible. The multi-scale encoder–decoder network helps to learn image features and reconstruct high-quality images.

Our work provides a solid foundation for state-of-the-art imaging systems. It can solve the problem that the imaging system is large and not easy to carry, and it is conducive to the miniaturization and integration of the imaging system. This work can be used in many fields such as smartphones, VR/AR glasses, and surgery. However, this paper still has some limitations. There is a deviation between simulation and actual manufacture, and the phase of the metalens obtained by simulation is different from that obtained by actual manufacture. Phase error and psf error should be considered in future research work. At the same time, the effect of the incident angle of the light on the image of the metalens is not considered in this paper. In the future, we will study the phase of the metalens in the case of oblique incidence and how to build a high-quality imaging system.

## 4. Conclusions

In this study, a high-quality imaging system that jointly optimizes hardware and recovery algorithms based on metalens phase factors and network parameters is proposed. Our approach incorporates hyperbolic and polynomial phases within the metalens phase profile, introducing optimization coefficients to enhance metalens performance. Furthermore, we employed an end-to-end neural network to jointly optimize the polynomial factors of the metalens phase and the network parameters, achieving effective chromatic aberration and high-resolution image reconstruction. Compared with the hyperbolic phase and cubic phase methods separately, the method used in this paper yields superior image quality, with the average PSNR reaching 28.53 and the average SSIM reaching 0.83. These results underscore the effectiveness of our integrated hardware and algorithmic optimization strategy. The simulation results demonstrate the exceptional imaging performance of our system, underscoring its potential to advance the miniaturization and integration of imaging systems. In the future, we will study both full-color and varifocal imaging systems. Our approach holds promise for a wide range of applications, including medical imaging, remote sensing, and consumer electronics.

## Figures and Tables

**Figure 1 nanomaterials-14-00715-f001:**
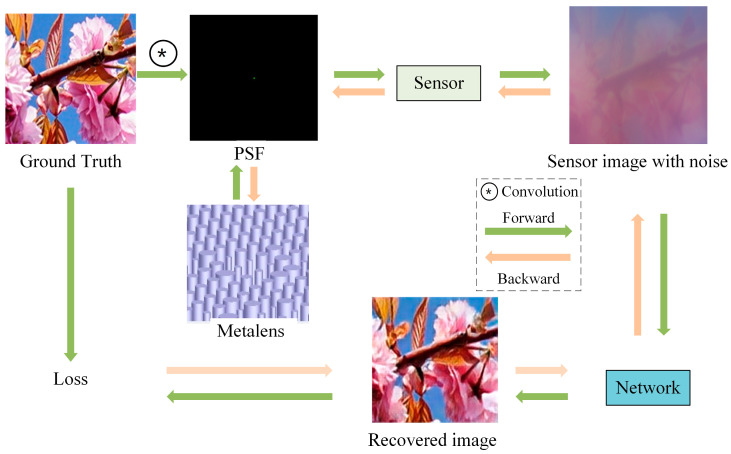
The imaging reconstruction system.

**Figure 2 nanomaterials-14-00715-f002:**
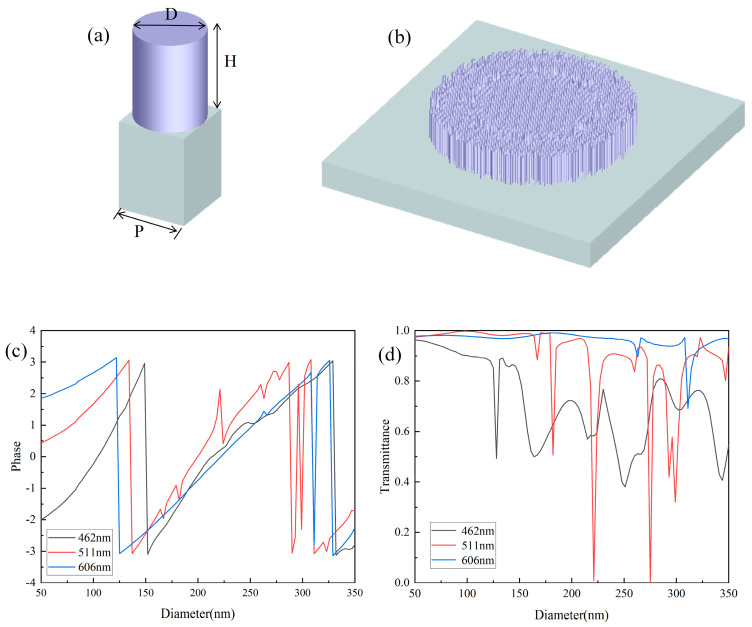
(**a**) Three-dimensional view of a unit cell; (**b**) top view of the unit cells; (**c**) the phase of unit cells; and (**d**) the transmittance of unit cells.

**Figure 3 nanomaterials-14-00715-f003:**
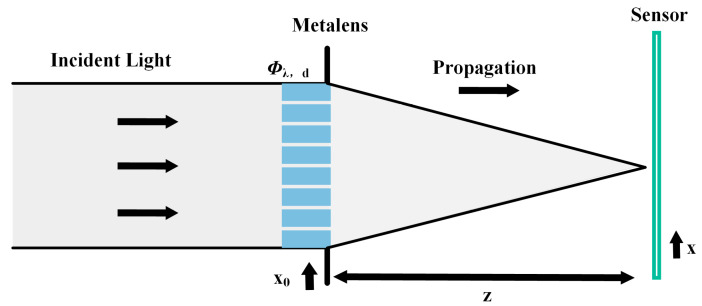
The metalens diffraction imaging schematic scheme.

**Figure 4 nanomaterials-14-00715-f004:**
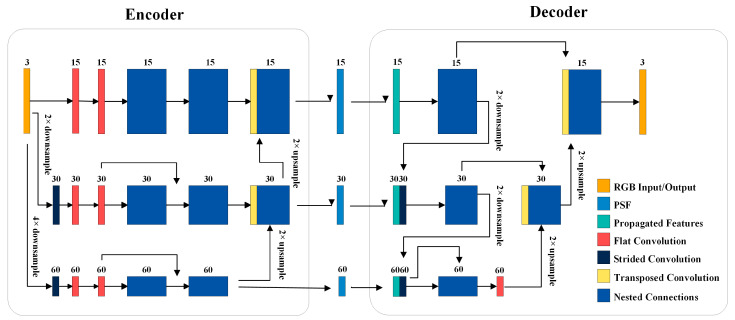
The multi-scale encoder–decoder based on a convolutional neural network.

**Figure 5 nanomaterials-14-00715-f005:**
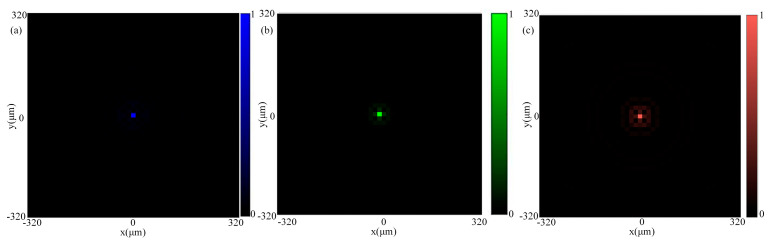
(**a**) The relative intensity of the PSF of 462 nm; (**b**) the relative intensity of the PSF of 511 nm; and (**c**) the relative intensity of the PSF of 606 nm.

**Figure 6 nanomaterials-14-00715-f006:**
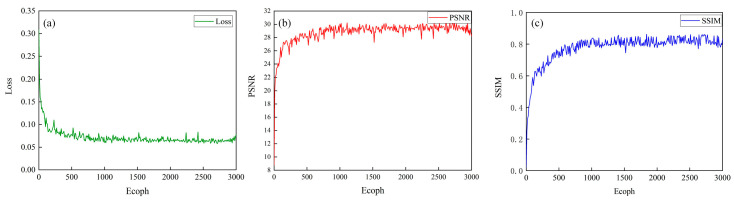
(**a**) Schematic diagram of loss function changes; (**b**) schematic diagram of PSNR changes; and (**c**) schematic diagram of SSIM changes.

**Figure 7 nanomaterials-14-00715-f007:**
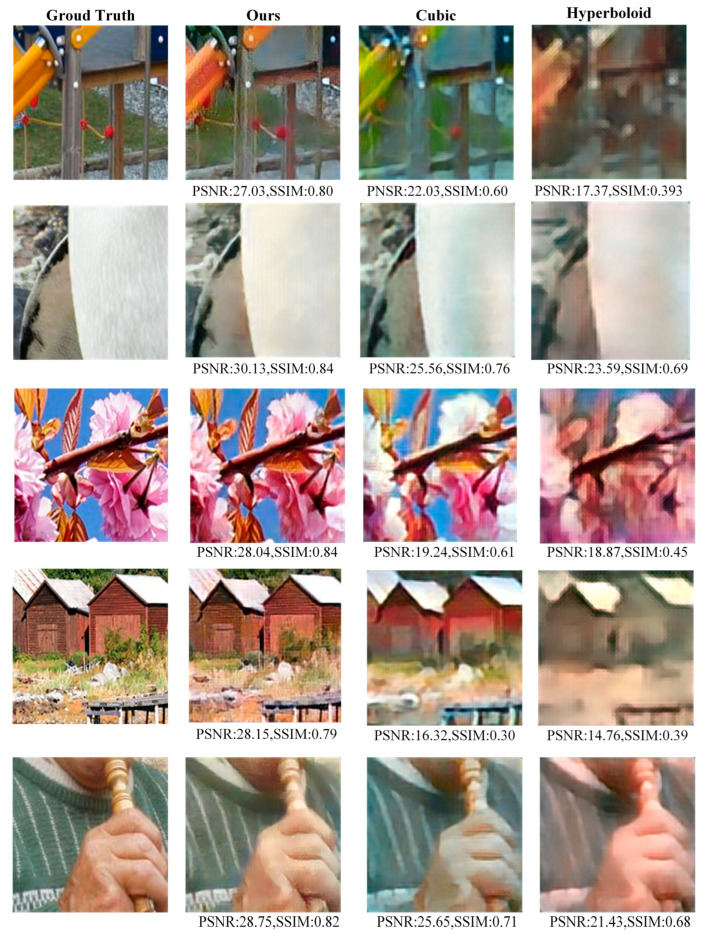
The imaging reconstruction system results for different metalens phases.

**Table 1 nanomaterials-14-00715-t001:** Average values of the PSNR and SSIM for the images.

	PSNR	SSIM
Ours	28.53	0.83
Cubic [21]	22.15	0.61
Hyperboloid [6]	17.54	0.52

## Data Availability

The data that support the plots within this paper and the other findings of this study are available from the corresponding authors upon reasonable request.

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
