# Peer review of "Full-Color Imaging System Based on the Joint Integration of a Metalens and Neural Network"

_nanomaterials, 2024, doi:10.3390/nano14080715_

Round 1
Reviewer 1 Report
Comments and Suggestions for Authors
Below are a few of my comments. Please address them accordingly.
The abstract requires thorough revision to enhance clarity and coherence. During the revision process, special attention should be given to highlighting the key findings.
Various recently published important papers on neural network applications are missing from the manuscript. Please discuss the following publications in your manuscript (J. Hazard. Mater. 462 (2024) 132773; J. Mater. Chem. A., 11 (2023) 9009-0918; Sep. Purif. Technol. 326 (2023) 124891). Besides, Please clearly highlight the key objectives of this study.
Figure 1 could be revised.
The result and discussion section is too small and does not reflect the entire methodology. please revise properly.
Comments on the Quality of English Language
A lot of English grammar and typo errors were seen throughout the manuscript. please revise carefully.
Author Response
Response to Reviewer Comments
Dear reviewer,
Thank you very much for your professional review. We sincerely appreciate your careful attention to detail and the valuable suggestions you have provided. Your feedback has greatly contributed to the quality of this manuscript. Below are our responses to each of your comments:
Point 1: The abstract requires thorough revision to enhance clarity and coherence. During the revision process, special attention should be given to highlighting the key findings.
Response 1: Thanks for your professional guidance. We revised the abstract to make it clearer and easier to understand.
Point 2: Various recently published important papers on neural network applications are missing from the manuscript. Please discuss the following publications in your manuscript (J. Hazard. Mater. 462 (2024) 132773; J. Mater. Chem. A., 11 (2023) 9009-0918; Sep. Purif. Technol. 326 (2023) 124891). Besides, Please clearly highlight the key objectives of this study.
Response 2: Thank you very much for your thorough review. We have cited these three articles, which can be seen in References 13-15. We have added introduction section and described this work in detail in the last paragraph of the section 1.
Point 3: Figure 1 could be revised.
Response 3: Thank you for your suggestions. We have modified Figure 1 to make it clearer.
Point 4: The result and discussion section is too small and does not reflect the entire methodology. please revise properly.
Response 4: Thank you for your thorough review. In the section 3, we have added the description of the network parameters and the analysis of the results, including the performance of PSF, Loss, PSNR, SSIM, etc.
Point 5: A lot of English grammar and typo errors were seen throughout the manuscript. please revise carefully.
Response 5: Thank you for your thorough review. We have checked the full text and made corrections where there are grammatical errors.
Finally, we would like to express our heartfelt gratitude for your advice and recommendations, which helped us improve our article.
Best wishes.
2024.04.09

Reviewer 2 Report
Comments and Suggestions for Authors
The authors presented a good work. However, there are some comments which should be addressed.
Major Comments:
The manuscript lacks a clear and concise explanation of the novelty and significance of the proposed approach in comparison to existing techniques. The introduction section should be strengthened to highlight the gaps in the current state-of-the-art and how this work addresses them.
The theoretical analyses section (Section 2) is dense and requires better organization and clarity. The explanations of the metalens design, PSF calculation, network architecture, and loss definition should be more explicit and easy to follow.
The authors should provide more details on the training process, including the number of iterations, batch size, hardware specifications, and training time. Additionally, the hyperparameter settings for the neural network should be clearly stated.
The results section (Section 3) is lacking in quantitative analyses. The authors should provide more comprehensive numerical results, including PSNR and SSIM values for a diverse set of test images, along with statistical analyses.
The manuscript would benefit from a more in-depth discussion on the limitations and potential drawbacks of the proposed approach, as well as suggestions for future improvements.
The quality of the figures, especially Figure 5, needs to be improved. Higher resolution and clearer images are necessary for better visualization and comparison.
The authors should provide more information on the dataset used for training and testing, including its source, size, and diversity.
The manuscript lacks a clear description of the experimental setup and validation methodology. The authors should describe in detail how the ground truth images were obtained and how the sensor images were simulated or captured.
Minor Comments:
Some of the acronyms used in the manuscript are not defined, making it difficult for readers to understand them. The authors should ensure that all acronyms are properly defined upon their first usage.
The writing style could be improved for better clarity and readability. There are instances of grammatical errors and awkward sentence structures throughout the manuscript.
The citation style is inconsistent, and some references are missing important information such as page numbers or publication years.
The authors should consider adding a section or subsection dedicated to discussing the potential applications and impact of their proposed approach.
The conclusion section could be expanded to provide a more comprehensive summary of the work and its implications, as well as suggestions for future research directions.
Author Response
Response to Reviewer Comments
Daer reviewer,
Thank you very much for your professional review. We sincerely appreciate your careful attention to detail and the valuable suggestions you have provided. Your feedback has greatly contributed to the quality of this manuscript. Below are our responses to each of your comments:
Point 1: The manuscript lacks a clear and concise explanation of the novelty and significance of the proposed approach in comparison to existing techniques. The introduction section should be strengthened to highlight the gaps in the current state-of-the-art and how this work addresses them.
Response 1: Thanks for your professional guidance. We have expanded on the introduction. As shown in the third and fourth paragraphs of the section 1, the current research status and shortcomings are summarized. In addition, the last summary of the first paragraph is expanded, and the main research direction and innovation points of the paper are put forward.
Point 2: The theoretical analyses section (Section 2) is dense and requires better organization and clarity. The explanations of the metalens design, PSF calculation, network architecture, and loss definition should be more explicit and easy to follow.
Response 2: Thank you very much for your thorough review. In the second section, we have expanded on the topics of metalens design, PSF introduction, neural network overview, and loss function definition. In the metalens design section, we have incorporated content on the principles of propagation phase and unit structure design. For the point spread function calculation, we have provided an introduction to the point spread function. In the neural network section, we have divided the network introduction into encoder and decoder parts. Additionally, in the definition of the loss function, we have included an introduction to the perceptual loss function.
Additionally, we have carefully reviewed the references cited in the first section to ensure the coherence and validity of our arguments.
Point 3: The authors should provide more details on the training process, including the number of iterations, batch size, hardware specifications, and training time. Additionally, the hyperparameter settings for the neural network should be clearly stated.
Response 3: Thank you for your suggestions. In the third section, we have added a subsection on experimental parameters and network parameter settings. We have provided detailed explanations of the neural network training process, including the number of training iterations, batch size, hardware specifications, training time and settings for hyperparameters.
Point 4: The results section (Section 3) is lacking in quantitative analyses. The authors should provide more comprehensive numerical results, including PSNR and SSIM values for a diverse set of test images, along with statistical analyses.
Response 4: Thank you for your thorough review. In the third section, we have included schematic diagrams of the trained PSF, as well as graphs showing the changes in loss, PSNR, and SSIM over the iteration count. These additions aim to enhance the clarity of our results.
Point 5: The manuscript would benefit from a more in-depth discussion on the limitations and potential drawbacks of the proposed approach, as well as suggestions for future improvements.
Response 5: We have added section 3.3, which discusses the limitations of the proposed approach, as well as suggestions for future improvements.
Point 6: The quality of the figures, especially Figure 5, needs to be improved. Higher resolution and clearer images are necessary for better visualization and comparison.
Response 6: Thank you for your thorough review. We carefully reviewed the images and resubmitted them to make sure the resolution met the magazine's requirements.
Point 7: The authors should provide more information on the dataset used for training and testing, including its source, size, and diversity.
Response 7: Thank you for your suggestion. In this paper, both our training and testing sets are sourced from the DIV2K dataset. In the experimental parameter setting subsection of the section3, we have added an introduction to the DIV2K dataset, including its diversity and size.
Point 8: The manuscript lacks a clear description of the experimental setup and validation methodology. The authors should describe in detail how the ground truth images were obtained and how the sensor images were simulated or captured.
Response 8: Our ground truth are indeed sourced from the DIV2K dataset. We have introduced the acquisition of sensor images in the PSF calculation section, which involves convolving real images with PSF and adding some noise, as shown in eq6. This process is implemented using Python code.
Minor Comments:
Point 9: Some of the acronyms used in the manuscript are not defined, making it difficult for readers to understand them. The authors should ensure that all acronyms are properly defined upon their first usage.
Response 9: Thank you for your correction. We have carefully reviewed the entire manuscript and ensured that the all acronyms are defined the first time they appear.
Point 10: The writing style could be improved for better clarity and readability. There are instances of grammatical errors and awkward sentence structures throughout the manuscript.
Response 10: Thank you for your feedback. We have thoroughly read through the entire manuscript, making modifications to certain sentences as necessary. Additionally, we have carefully checked the document for any grammar errors that may have occurred during the revision process.
Point 11: The citation style is inconsistent, and some references are missing important information such as page numbers or publication years.
Response 11: Thank you for your feedback. We have checked the format of the references and ensured that they adhere to the formatting requirements specified by the journal.
Point 12: The authors should consider adding a section or subsection dedicated to discussing the potential applications and impact of their proposed approach.
Response 12: Thank you for your feedback. We have added section 3.3, which summarizes the approach in this article and describes its advantages and potential applications.
Point 13: The conclusion section could be expanded to provide a more comprehensive summary of the work and its implications, as well as suggestions for future research directions.
Response 13: Thank you for your feedback. In the conclusion section, we have expanded upon our discussion, summarizing the main methods and results of this paper. Additionally, we have provided an overview of potential future research directions and future research directions for this method.
Best wishes.
2024.04.09

Reviewer 3 Report
Comments and Suggestions for Authors
Comment to the authors
I proceeded to analyze the manuscript entitled:
Full color imaging system based on the jointly integration of metalens and neural network
Written by: Shuling Hu, Ruixue Shi, Bin Wang, Yuan Wei, Binzhi Qi, Peng Zhou
The manuscript deals with a high-performance imaging system that uses metalens and image recovery algorithms together. It explores different metalens designs and uses a neural network to achieve superior image quality, including correcting chromatic aberration and achieving high resolution. The authors claim that this method outperforms traditional approaches, with simulations showing significant improvement in image quality using the peak signal to noise ratio (PSNR) of 28.53 and structure similarity(SSIM) of 0.83. The authors suggest that their results paves the way for smaller and more integrated imaging systems.
The topic is, in my opinion, interesting and using Artificial Neural Networks in data processing is quite actual. The figures are suggestive and support the statements. References are in proper amount and indicate that the authors are well aware of what has been published on the subject they are writing about. The article is well written, using good English, in my opinion, but a careful check would improve it. The content of the article sustains the Conclusion.
Moving to details, I found mention a few parts that, in my opinion, require improvement and additional clarification, as indicated on each item, and they are mentioned below.
-pg1 Line 6574 and Fig1 are more suited in Section 2. Instead introduce a brief linking sentence to section 2.
-pg3 eq.(1): verify and correct if the case. The second term, as written, can be written with the round bracket as a factor in front of the sum.
-Carefully verify all the equations.
-Briefly explain the terms PSF, metalens, cubic phase and hyperboloid phase before using them, as you submitted the manuscript to Nanomaterials, not to an optics engineering journal, therefore not all the readers are familiar with these concepts.
-page 4: .2. PSF caculate – isn’t is calcualtaion more appropriate?
-page 5: give more details on the ANN architecture
Comments on the Quality of English Language
Carefully check the manuscript before resubmitting the revised version
Author Response
Response to Reviewer Comments
Dear reviewer,
Thank you very much for your professional review. We sincerely appreciate your careful attention to detail and the valuable suggestions you have provided. Your feedback has greatly contributed to the quality of this manuscript. Below are our responses to each of your comments:
Point 1: pg1 Line 6574 and Fig1 are more suited in Section 2. Instead introduce a brief linking sentence to section 2.
Response 1: Thanks for your professional guidance. We have carefully considered the structure of the full text and decided that Figure 1 would be more appropriate in the section 1. But we have expanded our introduction to section 2.
Point 2: pg3 eq.(1): verify and correct if the case. The second term, as written, can be written with the round bracket as a factor in front of the sum.
Response 2: Thank you very much for your thorough review and for correcting the error in our Equation 1. We have now modified eq.(1) to the correct form.
Point 3: Carefully verify all the equations.
Response 3: Thank you for your suggestions. We have thoroughly reviewed the entire manuscript and made corrections to eq.(1) and (8) as advised.
Point 4: Briefly explain the terms PSF, metalens, cubic phase and hyperboloid phase before using them, as you submitted the manuscript to Nanomaterials, not to an optics engineering journal, therefore not all the readers are familiar with these concepts.
Response 4: Thank you for your thorough review. In the section 2, we have provided detailed descriptions of the metalens and the psf. Additionally, in the results section (Section 3), we have included the formulas for hyperbolic and cubic phase profiles.
Point 5: PSF caculate – isn’t is calcualtaion more appropriate?
Response 5: Thank you for your suggestion. We have now updated the title to "PSF Calculation.
Point 6: give more details on the ANN architecture.
Response 6: Thank you for your suggestion. In the section on network architecture, we have provided a detailed description of the encoder and decoder network.
Finally, we would like to express our heartfelt gratitude for your advice and recommendations, which helped us improve our article.
Best wishes.
2024.04.09

Round 2
Reviewer 1 Report
Comments and Suggestions for Authors
the authors have made sufficient improvement. thus, it can be accepted.
Comments on the Quality of English Language
it is acceptable
Reviewer 2 Report
Comments and Suggestions for Authors
The Authors have addressed all the comments.
Reviewer 3 Report
Comments and Suggestions for Authors
I proceeded to re-analyze the manuscript entitled:
Full color imaging system based on the jointly integration of metalens and neural network
Written by: Shuling Hu, Ruixue Shi, Bin Wang, Yuan Wei, Binzhi Qi, Peng Zhou
The authors answered the points I raised in the revised version of the manuscript.
I have no further comment.